

# Beaver dam influences on streamflow hydraulic properties and thermal regimes

Milada Majerova [1], Bethany T. Neilson [1], Brett B. Roper [2,3]

[1] Utah Water Research Laboratory, Department of Civil and Environmental Engineering, Utah State University, 8200 Old Main Hill, Logan, Utah, 84322-8200, United States

[2] Department of Watershed Sciences, Utah State University, 8200 Old Main Hill, Logan, Utah 84322-8200, United States

[3] Fish and Aquatic Ecology Unit, U.S. Forest Service, 860 North 1200 East, Logan, Utah 84321, United States

*Correspondence to:* M. Majerova (milada.majerova@gmail.com) and B.T. Neilson (bethany.neilson@usu.edu)

**Abstract.** Beaver dams alter channel hydraulics which in turn change the geomorphic templates of streams. Variability in geomorphic units, the building blocks of stream systems, and water temperature, critical to stream ecological function, define habitat heterogeneity and availability. While prior research has shown the impact of beaver dams on stream hydraulics, geomorphic template, or temperature, the connections or feedbacks between these habitat measures are not well understood. This has left questions regarding relationships between temperature variability at different spatial scales to hydraulic properties such as flow depth and velocity that are dependent on the geomorphology. We combine detailed predicted hydraulic properties, field based maps with an additional classification scheme of geomorphic units, and detailed water temperature observations throughout a study reach to demonstrate the relationship between these factors at different spatial scales (reach, beaver dam complexes, and geomorphic units). Over a three week, low flow period we found temperature to vary 2 °C between the upstream and downstream extents of the reach with a net warming of 1 °C during the day and a net cooling of 0.5 °C at night. At the beaver dam complex scale, net warming of 1.15 °C occurred during the day with variable cooling at night. Regardless of limited temperature changes at these larger scales, the temperaure variability in a beaver dam complex reached up to 10.5 °C due to the diversity of geomorphic units within the complex. At the geomorphic unit scale, the highly altered flow velocity and depth distributions within primary units provide an explanation of the temperature variability within the dam complex. Riffles, with the greatest velocity variability and least depth variability, have the smallest temperature variability and range. The lowest velocity variability occurred within margins, pools, and backwaters which exhibit the widest temperature ranges, but range from shallow to deep. Overall, the predicted flow hydraulic properties for different geomorphic units suggest that velocity is the primary factor in determining the variability of water temperature. However, water depth can also play a role as it impacts warming patterns and can dictate thermal stratification. These findings begin to link key attributes of different geomorphic units to thermal variability and illustrates the value of the geomorphic variability associated with the development of beaver dam complexes.

## 1 Introduction

The presence of beaver dams in streams changes channel hydraulics resulting in decreased flow velocities and increased flow depths within beaver ponds (Green and Westbrook, 2009, Nyssen et al., 2011, Westbrook et al., 2006)





and thus increases the hydraulic variability within stream reaches. Hydraulic diversity introduced to the system by beaver dams changes depositional and erosional processes of the stream (Pollock et al., 2007), resulting in a changed geomorphic template. Different geomorphic unit patterns, which are considered the building blocks of stream systems (Brierley, 1996), define the amount and variability of physical habitat along the streams (Brierley and Fryirs, 2013, Montgomery, 2001, Newson and Newson, 2000, Wheaton et al., 2010, Roegner et al., 2008). However, few studies have investigated the influences of beaver dams on the connections between channel hydraulics and the geomorphic template (e.g., Green and Westbrook, 2009, Pollock et al., 2007, Wheaton et al., 2004, Levine and Meyer, 2014, Stout et al., 2016).

Habitat availability and quality also require an understanding of water temperatures (Hickman and Raleigh, 1982, Dallas and Rivers-Moore, 2012, Allen, 1995). Water temperature is primarily dictated by climatic drivers (such as solar radiation, air temperature and wind speed), channel structure and complexity, groundwater influences, and riparian vegetation (Sullivan and Adams, 1991, Poole and Berman, 2001). Beaver dams and beaver activity can significantly alter many of these factors and change the relative importance of various heat transfer mechanisms (e.g., groundwater exchanges, Westhoff et al., 2007, Beschta, 1997, Keery et al., 2007, Hannah et al., 2004 ). Findings within the literature regarding the impacts of beaver dams on temperature have been contradictory. Some document longitudinal trends and overall increases in downstream temperature (Andersen, 2011, Margolis et al., 2001, Salyer, 1935, McRae and Edwards, 1994, Shetter and Whalls, 1955, Majerova et al., 2015). Others find longitudinal buffering of diel summer temperature extremes (Weber et al., 2017) or compare temperature across beaver ponds with increases in temperature below low-head beaver dams but cooling below high-head dams (Fuller and Peckarsky, 2011). At larger scales (~20 km), insignificant temperature changes have been observed due to beaver dam influences (Talabere, 2002). Majerova et al. (2015) highlighted the importance of spatial as well as temporal scales when examining the influences of beaver dams on temperature. They illustrated the role of individual beaver dams on cumulative downstream warming and/or cooling and demonstrated increased thermal variability after beaver colonization. Literature regarding the impacts of beaver dams on stream temperature in relation to fish are similarly inconsistent and few studies are based on in-situ measurements (Kemp et al., 2012, Gibson and Olden, 2014).

These individual studies all highlight that beaver dams impact stream hydraulics, geomorphic template, and water temperature. We also know stream temperature is influenced by channel complexity (longitudinally and laterally) and the associated variability in geomorphic units that creates habitat heterogeneity, often characterized by different temperature regimes (Dallas and Rivers-Moore, 2011, Poole and Berman, 2001, Schmadel et al., 2015). For example, pools can exhibit thermal stratification (Elliott, 2000, Nielsen et al., 1994, Tate et al., 2007), marginal areas can have higher temperatures (Clark et al., 1999), riffle temperatures may differ from pools (Nordlie and Arthur, 1981), backwaters can have higher summer maxima (Appleton, 1976, Harrison and Elsworth, 1958, Allanson, 1961), and small side channels can experience groundwater influences (Mosley, 1983). Regardless of such findings, the connections between stream hydraulics, geomorphic structure, and temperature are still not well understood. Many questions remain regarding our ability to relate different temperature responses at varied spatial scales (geomorphic



units, beaver dam complexes, or reaches) to detailed descriptions of hydraulic properties such as flow depth and
velocity.

To begin addressing the connections between habitat measures (channel hydraulics, geomorphic templates, and
temperature variability) and the influence of beaver dams and complexes, we first investigate the variability in
hydraulic properties throughout a reach influenced by beaver dams using a 2D hydraulic model. We compare
frequency distributions of depth and velocity at the reach and beaver dam complex scales. We then identify
geomorphic units based on classification tools and compare depth and velocity frequency distributions for geomorphic
units (pools, backwaters, margins, and riffles) and combine these results with temperature observations to establish
the role of hydraulic factors in dictating thermal responses at the beaver dam complex and geomorphic unit scales.
Finally, we illustrate the importance of measuring temperature responses at different spatial scales by comparing
temperature ranges at reach, beaver dam complex, and geomorphic unit scales.

**2 Site Description**
Curtis Creek, a tributary of the Blacksmith Fork River, is located in the northern Utah and drains a portion of the Bear
River Range. It is a first-order mountain stream with a snowmelt dominated hydrologic regime where runoff starts in
late April and continues until mid-June. The study reach, a 750 m long section of the stream, has a relatively steep
average slope of 0.035, supporting a streambed of coarse gravel to large cobble. The reach was part of Utah Division
of Wildlife Resources (UDWR) stream relocation project when in 2001, some segments of the channel (about 440 m
of stream length) were moved and reconstructed (Fig. 1, old channel). As a result, man-made boulder vortex weirs
were placed in the new channel with a meandering planform and the banks of the realigned channel were stabilized
with boulders, root wads, logs, and erosion control blankets. The riparian area surrounding the channel prior to and
following relocation was heavily grazed by elk and did not support woody riparian vegetation. Around 2005, grazing
pressure was lessened and the area was fenced (though some grazing was still allowed). This facilitated the modest
recovery of the riparian woody vegetation (Salix sp.) which attracted beaver and promoted beaver colonization in
early summer of 2009. Multiple dams with heights ranging from 0.5 to 1.3 m were built over the course of three years
resulting in dam density of 9.3 dam/km by year 2012 (Fig. 1). Beaver dams created ponded areas, promoted overbank
flooding, created new side channels, and reconnected the new channel with the old channel via damming. This
promoted channel-floodplain reconnection, especially in segments that were reconstructed and confined prior to
beaver colonization.

**3 Methods**

**3.1 Field data collection**
The study reach boundaries were set following previous studies (Schmadel et al., 2010, Majerova et al., 2015) and
represented a 750 m long reach (Fig. 1). An additional scale of interest is that of a beaver dam complex which includes
a beaver dam or a series of beaver dams that are close to each other, the beaver pond, a portion of the upstream channel,





and a portion of the downstream channel. Three beaver dam complexes were identified in the Curtis Creek study reach
(Fig. 1, black boxes).

Topographic data and water surface elevations were collected throughout the study reach using a differential rtkGPS
(Trimble® R8, Global Navigation Satellite System, Dayton, Ohio, USA). Main and side channel topography
resolution ranged from 1.0 to 4.5 points per m$^2$ with the resolution decreasing on the banks and floodplain (less than
or equal to 1 point per m$^2$). Water surface elevation data were collected longitudinally for base flow (0.19 m$^3$ s$^{-1}$, 2012)
and high flow conditions (0.93 m$^3$ s$^{-1}$, 2014) with point densities ranging from 1 point per 0.3 m of stream length to 1
point per 20 m of stream length. Discharge measurements were taken at both upstream and downstream boundaries
using Marsh McBirney Inc® Flo-Mate™ (Model 2000, Frederick, Maryland) at the time of WSEL survey.

Two different types of temperature sensors were deployed during the study period at two different spatial scales, the
geomorphic unit scale within the beaver dam complex and a reach scale. 25 HOBO Pro v2 temperature sensors (Onset
Computer Corporation, Cape Cod, MA) provided temperature data at the geomorphic unit scale in the beaver dam
complex #1 (Fig. 1) from September 6 to September 26, 2013 (Snow, 2014) at 5-minute intervals. In pools and deeper
backwater areas where stratification could be present, sensors were also placed in a vertical array throughout the water
column (up to three sensors in one location). In addition to this fine spatial resolution, 25 HOBO TidbiT v2 temperature
sensors were placed in the main channel throughout the study reach and were logging continuous water temperature
data every 10 minutes (Fig. 1).

**3.2 2D Model development**
To evaluate hydraulic properties, the open source software Delft3D 4.01 Suite/FLOW module was applied to our
study site. This multi-dimensional (2D or 3D) hydrodynamic model solves the shallow water equations derived from
the three dimensional Navier-Stokes equations for incompressible free surface flow. The equations used were
formulated in orthogonal curvilinear coordinates. Rectangular grids are considered a simplified form of a curvilinear
grid (Delft3D- FLOW User Manual, Version 3.15). Hydraulic calculations are grid based and thus model results are
presented in the grid cell form. ArcMap 10.2 was used to develop the Digital Elevation Model (DEM) from
topographic and bathymetric surveys which was later used to create a 0.4 x 0.4 m grid within Delft3D. Beaver dams
were included in the grid as part of the geometry. To ensure flow through the structures, the openings were created
manually to match the water surface elevations collected above the dams. Measured discharge was used for the
upstream boundary condition while measured water surface elevation was used for the downstream boundary. Initially,
high flows of 0.93 m$^3$ s$^{-1}$ were used for model calibration with later adjustment for low flow of 0.19 m$^3$ s$^{-1}$ to reflect
base flow conditions during summer. A Manning's $n$ value of 0.038 was determined via input parameter sensitivity
analysis and applied for the entire study reach for both low and high flow conditions. The same input parameter
analysis determined an eddy viscosity of 0.1 m$^2$ s$^{-1}$ to achieve the smallest RMSE values. A time step sensitivity
analysis showed that results were independent when a time step size of 0.0025 min or less was applied. Therefore, a
time step of 0.0025 min was chosen and used for all the simulations. While model results are available for both low



and high flow conditions, this work focused on low flow conditions when water temperature can be limiting. To evaluate model outputs at the different spatial scales, Delft3D output files from the low flow model results were processed to create depth and velocity distributions for the study reach and beaver dam complexes. Distributions were normalized by the total count for direct comparison of scales.

### 3.3 Geomorphic mapping

A spatially continuous map of the channel and floodplain identifying and describing individual geomorphic units was constructed from field observations that captured conditions at base flow in summer 2012 based on the approach described within Brierley and Fryirs (2013). Combining a field based delineation of geomorphic units and the DEM constructed from topographic and bathymetric surveys, we applied the classification scheme developed by Wheaton et al. (2015). This allowed for classification of margins, structural elements, and geomorphic units. Tiered classification of geomorphic units first considered stage height (tier 1), then shape (tier 2), and then morphology (tier 3). By overlaying the classified geomorphic units with the predicted velocity and depths, cells within the model domain were reclassified into 4 key geomorphic units (pool, backwater, channel margins, and riffle). Additionally, velocity and depth thresholds associated with each of these geomorphic units were established based on model predictions from each unit (Wyrick and Pasternack, 2014). The thresholds established for geomorphic units are: 1) riffles consisting of depths less than 0.4 m and velocities higher than 0.5 m s$^{-1}$, but including lower velocity, lateral cells so that riffles span the channel; 2) pools consisting of depths equal to or greater than 0.5 m and velocities below 0.5 m s$^{-1}$; 3) marginal areas consisting of depths less than 0.1 m, velocities that could not exceed 0.1 m s$^{-1}$, and usually span one to two cells from the water's edge; 4) backwater areas where velocities are less than 0.1 m s$^{-1}$ with varying depths, but had at least two adjacent cells to create a continuous surface. To quantify the variability in flow properties at different spatial scales, depth and velocity distributions were constructed for each of four geomorphic units at the reach and beaver dam complex scale.

### 3.4 Temperature data

To link hydraulic predictions and the geomorphic template to stream temperature, temperature data from September 2013 collected within the beaver dam complex #1 (Snow, 2014) were grouped by different geomorphic units. For comparison of the thermal responses at the beaver dam complex and study reach scales, temperature data from the extents of these scales were compared. Further, at the beaver dam complex scale (specifically beaver dam complex #1), a temperature range (minima and maxima) was constructed from the 35 sensors (at 25 locations) within the beaver dam complex to illustrate thermal variability by geomorphic unit for the same time period. Similarly, at the reach scale, temperature ranges captured by the 25 sensors from the main channel of the study reach were evaluated to determine the temperature variability at this scale.

### 4 Results

### 4.1 Comparison of computed and observed water surface elevations for 2D model


The calibrated 2D model generally under-predicted observed water surface elevations with the greatest differences
between computed and observed elevations being in the ponded areas. For the 564 comparison locations throughout
the study reach (SI Fig. 1), the average difference between the model and observed water surface elevation was -0.056
m, with an RMSE value of 0.078 m. Even though the model under-estimated water surface elevations in general,
computed values were higher 6 % of the time by 0.03 m on average.

**4.2 Geomorphic mapping**
By combining the field based delineation of geomorphic units and the DEM, a tier 3 classification scheme was applied
that resulted in a detailed map of the study reach and illustrates the influences of beaver dams on channel form and
structure (Fig. 2). During the study period, 7 beaver dams were located in the main channel and one was in the old
channel at the downstream extent of the reach (Fig 1, Fig. 2). Multiple additional small dams were present in the old
channel with herbaceous vegetation or smaller wooden branches being the primary building material. The most
upstream main channel dam breached a year prior to the mapping and degradation of the dam continued over the
following years. Beaver ponds represented about 33.5 % (1124 $m^2$) of the wetted channel area. Overflow channels
and beaver canals resulting from dam construction in the main channel created new flow paths that connected it to the
old channel and added 2020 $m^2$ of additional wetted area (Fig. 2). New gravel bars at the upstream end of the reach
were a result of the dam breach and previous sediment movement from upstream.

**4.3 Flow hydraulic properties**

**4.3.1 Study reach**
Flow depth and velocity calculated for each cell within the computational domain of the study reach ranged from 0.03
to 1.08 m and 0.001 to 2.8 m $s^{-1}$, respectively. The 0.03 m depth value is set in the model as a minimal depth threshold
and dictated when a computational cell was considered wet. The average depth and velocity for the entire study reach
was 0.23 m and 0.25 m $s^{-1}$, respectively. The depth frequency distribution for the reach was positively skewed with
majority of depths falling under 0.3 meters (Fig. 3A). The same trend was observed for the reach velocity distribution
where areas with low velocity (margins, backwaters) represented about 31 % of the channel.

Using the geomorphic unit classification (Fig. 2) and predictions of depth and velocity, pools, backwaters, margins,
and riffles represented 13, 21, 10, and 10 % of the entire reach computational domain, respectively.  These units
exhibited different flow properties with an average depth and velocity for pools, backwater, marginal areas, and riffles
being 0.66 m (0.50–1.08 m) and 0.11 m $s^{-1}$ (0.001–0.73 m $s^{-1}$), 0.38 m  (0.03–1.08 m) and 0.03 m/s (0–0.10 m $s^{-1}$),
and 0.06 m (0.03–0.10 m) and 0.03 m $s^{-1}$ (0–0.1 m $s^{-1}$), 0.13 m (0.03–0.4 m) and 0.64 m $s^{-1}$ (0.002–1.83 m $s^{-1}$),
respectively.

**4.3.2 Beaver dam complex**



Combined, the beaver dam complexes (#1- 3, Fig. 1) covered about 67 % of the entire study reach. Similar to the reach
scale results, the predicted flow depths ranged from 0.03 to 1.08 m with the average value of 0.27 m. The beaver dam
complexes include shallow margin and transitional zones as well as the deepest spots within the beaver ponds. These
areas also contained the lowest and often near zero velocities, with an average value of 0.175 m s$^{-1}$. Similar to the
study reach, the distributions were positively skewed for depth, however, there were greater percentages of shallow
marginal areas. The velocity distribution is similar in shape and magnitude to the reach scale (Fig. 3B).

Focusing on beaver dam complex #1 (Fig. 1), which covers about 25 % of the study reach, the pool, backwater,
marginal areas, and riffle geomorphic units represented 10, 37, 9, and 11 % respectively (Fig. 4). The frequency
distributions for these individual units show how depth and velocity vary significantly over finer spatial scales. Pool
depths ranged from 0.50 m to 0.88 m with an average depth of 0.62 m. The velocity distribution was positively skewed
with an average velocity of 0.09 m s$^{-1}$ (Fig. 3C). Backwaters had the largest depth range since they covered deep areas
as well as shallow zones, but averaged 0.32 m. Velocity distributions reflected the <0.1 m s$^{-1}$ threshold used to
delineate backwater units. Marginal areas included very shallow areas in the channel (<0.1 m) and thus had a positively
skewed velocity distribution that consisted of low values with many smaller than 0.01 m s$^{-1}$. The riffle depths resulted
in the most symmetrical distribution with a range from 0.03 to 0.33 m and an average of 0.14 m. Velocities were
highest in the riffles with values nearing 1.46 m s$^{-1}$.

**4.4 Water temperature**

**4.4.1 Study reach**
Temperatures through the study reach, as illustrated by observed temperature ranges (minima and maxima) over time
based on the 25 main channel sensors, show significant spatial variability over the three week study period in the Fall
of 2013 (Fig. 5A). The maximum difference at any time throughout the reach was nearly 2°C. However, if the
difference between the most upstream and downstream sensors (Fig. 5B) is only considered, the downstream net
warming is ~1 °C (positive values) during the day and net cooling is 0.5 °C during the night (negative values).

**4.4.2 Beaver dam complex #1 and its geomorphic units**
At the finer scale of the beaver dam complex, similar to the reach scale, the pond warmed by about 1.15 °C (Fig. 5D)
during the day. However, the cooling effect at night is not present as often and responds differently than the reach
scale (Fig. 5B, 5D) in that the temperature reaches its maxima sooner in the day. The temperature decreased more
rapidly after the daily peak and the downstream cooling is observed earlier (Fig. 5B, 5D). The temperature sensors
placed throughout the beaver dam complex #1 (Fig. 1) demonstrate a wider range of temperatures with maximum
differences between temperature minima and maxima approaching 10.5 °C at times. To investigate this temperature
variability at the finer geomorphic unit scale, these same sensors were grouped by geomorphic units within the beaver
dam complex (Fig. 4). The temperature variability within units, as represented by maximum values minus minumum
values observed accross all sensors within a gemorphic unit classification over time (Fig. 6), show that backwaters





have the greatest variability with temperature ranges reaching 10.5 °C. Margins have the second highest varibility (5.6
°C), followed by pools with 4.1 °C (Fig. 6, SI Fig. 2).  No vertical thermal stratification was found in the pools in the
main portion of the beaver pond and only small temperature differences were observed between vertical sensors within
this area. However, the pool in the backwater area (Fig. 4) experienced thermal stratification that continued into the
old channel (SI Fig. 3).  In addition to the different thermal regimes recorded vertically, time lags in temperature
maxima were also present and ranged from 3 hours between the surface and middle layer and between 3.5 and 5.5
hours in the middle and bottom layers (SI Fig. 2).  The thermal stratification was responsible for a large fraction of
the temperature range present within backwater (Fig. 6, and Fig. 7C) and also created the lowest and highest
temperatures among the four geomorphic units (Fig. 7C). Margins also exhibited wide temperature ranges but were
similar to those found within pools. As expected, riffles were the least thermally variable with the riffle above and
below the pond showing similar temperature ranges and averages. However, when comparing the riffles above and
below, the difference in temperature reached up to 1.4 °C and illustrated the warming effect of the pond (Fig. 7C, SI
Fig. 4).

**4.5 Connecting flow hydraulic predictions, geomorphic units and stream temperature**
The flow depth and velocity ranges constructed from the model hydraulic predictions showed that backwater had the
largest depth range (0.03–0.88 m) and a relatively small velocity range (0.0–0.1 m s$^{-1}$), but had the greatest thermal
variability. At the same time, margins had the smallest depth (0.03–0.1 m) and velocity range (0.0–0.1 m s$^{-1}$), but still
had relatively large temperature variability. Pools had the second largest depth range (0.5–0.88 m) and the velocity
range (0.0–0.55 m s$^{-1}$) was the third smallest, but the temperature variability was still high (Fig. 7). Riffles, with the
least thermal variability, had substantially larger velocity ranges and minimal depth ranges (Fig. 7).

**5 Discussion**

**5.1 Model Performance**
Use of a constant Manning's *n* for the entire model domain may have translated into a slight increase in the overall
RMSE value. Consistent with previous modeling efforts used for habitat analysis Jowett and Duncan, 2012, however,
the sensitivity analysis showed that Manning's *n* does not notably impact computed water surface elevations (SI Fig.
6, SI Fig. 7, SI Table 1). This suggests that water surface elevations were mainly influenced by bed topography and
the derived computational mesh as well as chosen eddy viscosity parameter. However, another possible error source
could be the treatment of beaver dams and flow through them within the modeling. Flow through dams that were part
of the channel topography was ensured via openings in the dam in an effort to mimic observed water surface elevations
immediately upstream of the structure. This may have led to computational inaccuracies around the dam structures
themselves. Different methods for handling flow through the dams may improve overall model accuracy.

**5.2 Geomorphic mapping**
The detailed classification map of the study reach illustrates the impacts of beaver dam development through the





diversity of geomorphic units, channel adjustments, and new flow paths throughout the reach (Fig. 2). By combining
field based observations with the tier classification map, the in-channel geomorphic unit delineations were more
confidently identified and provided the baseline information for further hydraulic analyses. Additionally,
temperature sensors were generally placed in the center of the units so small deviation in the boundary delineations
could influence depth and velocity frequency distributions, but would not significantly alter the identified thermal
variability within these units.

**5.3 Flow depth and velocity frequency distributions**

Depth and velocity distributions for the reach and beaver dam complexes follow similar trends primarily because
beaver dam complexes comprise a significant portion of the reach. When considering the geomorphic units within
beaver dam complex #1, the depth and velocity distributions clearly differ from the reach and beaver dam complex
scales (Fig. 3). Previous efforts have shown pools to have the widest velocity and depth distributions and include more
diverse microhabitat (Rosenfeld et al., 2011). In our study, pools had the second widest depth distribution (Fig. 3, Fig.
7). Backwater areas, which are created when beaver dams are constructed and have not typically been separated out
in previous studies, demonstrated the widest range of depths in our study. Both, pools and backwaters cover deep and
low velocity areas of the channel and were mainly a result of beaver dam construction. Stout et al. (2016) made a
comparison of the same study reach both with and without beaver dams. They concluded that there was a 50 % increase
in depths and 31 % decrease in velocities for this reach when the beaver dams are present. Although this comparison
is based on 1D model cross-sectional values that do not represent the geomorphic unit scale, it captures the longitudinal
heterogeneity of the hydraulics.

**5.4 Hydraulic properties, geomorphic units, and thermal variability**

The range of reach scale temperatures reflects variations within the reach (Fig. 5A) and highlights the warming effects
of a series of beaver complexes on longitudinal stream temperature patterns (Fig. 5B). The temperature sensors placed
in the main channel flow experience vertically well mixed conditions and mostly have similar thermal regimes as
illustrated by the small temperature ranges observed over time (Fig. 5A), but are limited in density in gemorphically
complex areas (e.g., beaver dam complexes). However, temperature ranges constructed from the 35 sensors placed
throughout the dam complex and within many of the same geomorphic units illustrates that the spatial variability
throughout the complex approaches 10.5 °C. Similar to Majerova et al. (2015), these results highlight the importance
of the spatial scale and resolution at which the measurements and observations are made. The high density
measurements made within specific geomorphic units in the beaver dam complex (Fig. 6, 7, SI Fig. 2) better represent
the habitat diversity available for the various fish species and life stages. These wide temperature ranges represent the
influence of highly variable hydraulic properties (Fig. 3C) and complex hydraulic mixing patterns within different
geomorphic units that in turn influence dominant heat fluxes and thermal responses. This highlights that the variability
in geomorphic unit types within a beaver dam complex and the resulting, but highly interdependent, depth and velocity
distributions (Fig. 3C), are key in creating variable thermal regimes.





Geomorphic units within main flow of the pond and the riffles above and below the ponded area generally experience
vertically well mixed conditions and short residence times which result in similar temperature regimes (SI Fig. 4). The
lower velocity pools tend to experience greater temperature variability, but unlike other studies (Nielsen et al., 1994,
Tate et al., 2007, Elliott, 2000) no stratification was present. Clark et al. (1999) also observed limited stratification in
two rivers in the UK and attributed this to insufficient depths. Consistent with these findings, Butler and Hunt (2013)
observed stratification when depths were greater than 1 m. While both depth and velocities within pools are key to
quantifying thermal stratification, other factors such as dissolved organic carbon and turbidity must also be considered
(Merck and Neilson, 2012, Cory et al., 2015, Wang and Seyed-Yagoobi, 1994, Kirk, 1985). The lowest velocity areas
of the beaver pond have either the greatest depth (backwater) or the smallest depth (margins) and a range of ~3-22 °C
for backwater areas and ~5-19 °C for margins during the three week study period (Fig. 7). Within the backwater unit
near the boundary of the old channel, there is significant thermal stratification that contributes to the overall
temperature variability within the beaver dam complex (SI Fig. 3). The varied thermal responses within these units
are dependent on a number of factors, many of which can be tied back to hydraulic properties.

Thermal stratification within the backwater area is a result of low velocities that minimize lateral and vertical mixing
and increase residence times (SI Fig. 3). Additionally, rooted macrophyte growth created a shallow surface layer of
water that would warm significantly during the day due to solar radiation inputs, while the water beneath the thick
vegetation was shaded from solar influences. Combined with localized groundwater upwelling in this area, it is clear
how such strong thermal stratification could develop in relatively shallow areas. Similarly, Clark et al. (1999) observed
heating of the surface layer isolated by the vegetation in 40 study locations, out of which 24 locations experienced
more than 1 °C difference. They also observed time lags between the surface layer and main channel temperatures
and the differences in the timing of the peak was more pronounced than for the minimum daily values. In their study,
water temperature in the surface layer of the backwater area peaked on average 150 minutes earlier than in the main
channel. This differs from our observations where no time lag is present between the surface layer of the backwater
and main flow (SI Fig. 3). However, there was a time lag between the bottom layer and the main flow temperature
which reached up to 8 hours. These cool bottom layers can be extremely important refugia for fish survival in summer
months, especially in changing flow conditions over the last decade (Nielsen et al., 1994, Dallas and Rivers-Moore,
2012, Nielsen et al., 1994, Tate et al., 2007, SI Fig. 3, SI Fig. 5).

When considering temperature variability within the margins, low velocities and shallow depths translate into small
volume to surface area ratios and long residence times. As the surface area to volume ratio is increased, more energy
can be exchanged across the air-water interface area and with long residence times, the temperature of small parcels
of water can be significantly altered (e.g., Gu et al., 1998). In general, marginal areas are expected to have higher daily
temperatures (Appleton, 1976, Harrison and Elsworth, 1958, Allanson, 1961, Clark et al., 1999). We found these areas
had warmer temperatures during the day and a wide temperature range (Fig. 7). Energy gains during the day from the
sun and energy losses during the night due longwave radiative exchange and evaporation are generally the primary
causes of these large temperature changes. Others have found these areas to cool and heat differently than the main





channel (e.g., Rutherford et al., 1993), but these effects have also been found to vary during the day depending on the location, depth, and localized shading (Neilson et al., 2009). Further, Neilson et al. (2010) found these areas to be a heat source at night and a heat sink during a portion of the day. Regardless, these studies have focused on a more typical density of marginal areas that are lower than that observed within beaver dam complexes. Some preliminary modeling work to identify dominant heat fluxes within various portions of this beaver dam complex has shown that the thermal responses of many areas representing individual or combined geomorphic features are dominated by surface heat fluxes, radiation penetration of the water column, and the residence time (Snow, 2014). This further highlights the role of hydraulic properties and geomorphic templates on small scale temperature responses.

Beaver dams significantly contribute to spatial heterogeneity of hydraulic properties resulting in the changed geomorphic template of the stream that creates stream systems (Brierley, 1996) and defines the physical habitat diversity (Brierley and Fryirs, 2013, Montgomery, 2001, Newson and Newson, 2000, Wheaton et al., 2010, Roegner et al., 2008). In general, model predictions of flow hydraulics within different geomorphic units and the associated temperature variability illustrate the dominant role of velocity in thermal responses as it more directly represents residence time distributions. The temperature variability within marginal areas, backwater, and pools illustrate this point well (Fig. 7C). Overall, when assessing geomorphic units and predicted hydraulic properties, the variability in temperature regimes can generally be explained. While the localized and site specific conditions (e.g., shading and groundwater exchanges) can create many thermal anomalies, identification of geomorphic units and the associated hydraulic properties will allow one to anticipate the potential thermal variability within each unit. These estimates can be based on velocity distributions, but depth is still important due it providing a volume surrogate that represents the potential for thermal buffering. Regardless, it is important to remember that absolute temperatures in streams are only partially dictated by hydraulic properties as many other factors must be considered (e.g., surface heat fluxes, groundwater exchanges, shading, water chemistry, aquatic vegetation). In areas of beaver dam complex development, it is clear that the dams increase the development of varied geomorphic units that correspond with lower velocities, higher residence times, and significant depth and temperature variability which all serve to diversify aquatic habitat. The thermal and physical diversity of conditions found within beaver dam complexes have been shown to improve trout growth (Sigourney et al., 2006 ) and suggest that stream sections with beaver dams will likely increase overall trout production (Gard, 1961) even if total counts are not higher. Therefore, the widespread presence of beaver dam complexes in a watershed would likely only positively affect trout population dynamics.

**6 Conclusion**

This study relates stream hydraulics and the geomorphic template of a stream impacted by beaver dams to stream temperature; an important indicator of habitat availability and quality. Using predicted hydraulic properties, detailed field observations of geomorphic units, and water temperature measurements, we demonstrate that geomorphic units within beaver dam complexes exhibit highly unique thermal responses in part due to the variability in flow velocities and depths. Velocity plays a more dominant role in temperature distributions as it provides a more accurate indicator of residence time. While geomorphic units within main flow of the river generally experience vertically well mixed





conditions and uniform temperatures, the lower velocity pools, backwaters and margins tend to experience greater
temperature variability. Observed thermal stratification in the backwaters was attributed to low velocities as well as
macrophyte growth and local groundwater inputs in the area. Low velocities and shallow depths of marginal areas
translate into small volume to surface area ratios and long residence times resulting in wide daily variations in
temperature.

This study also illustrates the importance of scale by comparing temperature responses across reach and beaver dam
complex scales. We observed the warming effects of multiple beaver dam complexes on longitudinal stream
temperature as captured by the 2 °C within reach temperature differences. In contrast, when temperature is measured
at smaller spatial scales, temperature differences within individual geomorphic units reached up to 10.5 °C within a
beaver dam complex. This wide temperature range illustrates the influence of highly variable depth and velocity
distributions and complex hydraulic mixing patterns within different geomorphic units.

Beaver dams significantly contribute to spatial heterogeneity of hydraulic properties resulting in a changed
geomorphic template of streams. We demonstrated this imposed variability through predicted spatial distributions of
hydraulic properties within a reach with multiple beaver dam complexes containing diverse geomorphic units. We
additionally illustrated how changing hydraulics influenced the variability of thermal responses and provide insight
regarding links in geomorphic changes and various habitat diversity measures.

**Acknowledgements**
This research was primarily funded by the Utah Water Research Laboratory and funding provided as part of a
cooperative agreement with the U.S. Forest Service's Watershed, Fish, Wildlife, Air, and Rare Plants staff group. The
authors would additionally like to thank the Utah Division of Wildlife Resources for facilitating this research and the
numerous field crew members for their help with data collection. We are also thankful to Joe Wheaton for feedback
and discussion in early stages of this paper. The support and resources from Center for High Performance Computing
at the University of Utah are gratefully acknowledged. Namely, help of Wim R. Cardoen with providing technical
guidance is appreciated.

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





Figure 1 Curtis Creek study reach and beaver dam complexes (black boxes, #1-#3) showing beaver dams with
associated beaver ponds, reach and fine beaver dam complex #1 temperature sensors, and pressure transducers at the
upstream and downstream end (red squares). The old channel is represented by blue dashed line. Water depth
displayed was created from bathymetric data and observed water surface elevation data. It captures different depths
within the main channel but also illustrates simplified water surface area in the study reach. Flow is from right to left
(A). Spatial scale scheme is shown in (B).

Figure 2 Tier 3 classification (Wheaton et al., 2015) of Curtis Creek study reach showing margins, structural
elements, and specific geomorphic units in and out-of-channel. Flow is from right to left.

Figure 3 Normalized depth and velocity distributions for the study reach (A), beaver dam complexes in the reach (B,
black boxes in Figure 1), and beaver dam complex #1 with its four geomorphic units (C, Fig. 1) constructed from 2D
model predictions.

Figure 4 Tier 3 classification (Wheaton et al., 2015) of the study reach showing beaver dam complex #1 in detail
(B). Temperature sensors were placed throughout the complex to investigate how temperature defers among the
individual geomorphic units, and above and below the beaver pond.

Figure 5 Temperature ranges at the study reach and beaver dam complex scales. A) Temperature ranges throughout
the main channel of study reach constructed from 25 temperature sensors placed longitudinally (Fig. 1).
Temperature at the upstream and downstream end of the reach illustrates a small overall warming effect at the
downstream end. Positive values in temperature differences (B, grey line) represent warming and negative values
represent cooling effect at the downstream end of the reach. C) Temperature range within the beaver dam complex
#1 from 35 sensors placed in different geomorphic units throughout the complex (Fig. 1, Fig. 3). Temperatures
above and below the beaver pond capture pond influences on downstream temperatures with temperature difference
(grey line) showing either warming (positive values) or cooling (negative values) (D).

Figure 6 Temperature difference (maxima minus minima) for individual geomorphic units within beaver dam
complex #1 for a period of twenty days during base flow conditions in September. Lines represent temperature
variation within pools (solid light blue), backwater (dashed dark blue), and marginal areas (dotted yellow). The
dashed red line illustrates influence of the beaver pond by showing differences between temperature below and
above the pond where positive values mean downstream warming and negative values mean downstream cooling
effect.



Figure 7 Model hydraulic predictions of depth and velocity as ranges of values for individual geomorphic units
within the beaver dam complex (A,B). Temperature ranges for same geomorphic units in the beaver dam complex
showing temperature variability for base flow conditions where n = the number of temperature sensors within a
geomorphic unit classification (C).







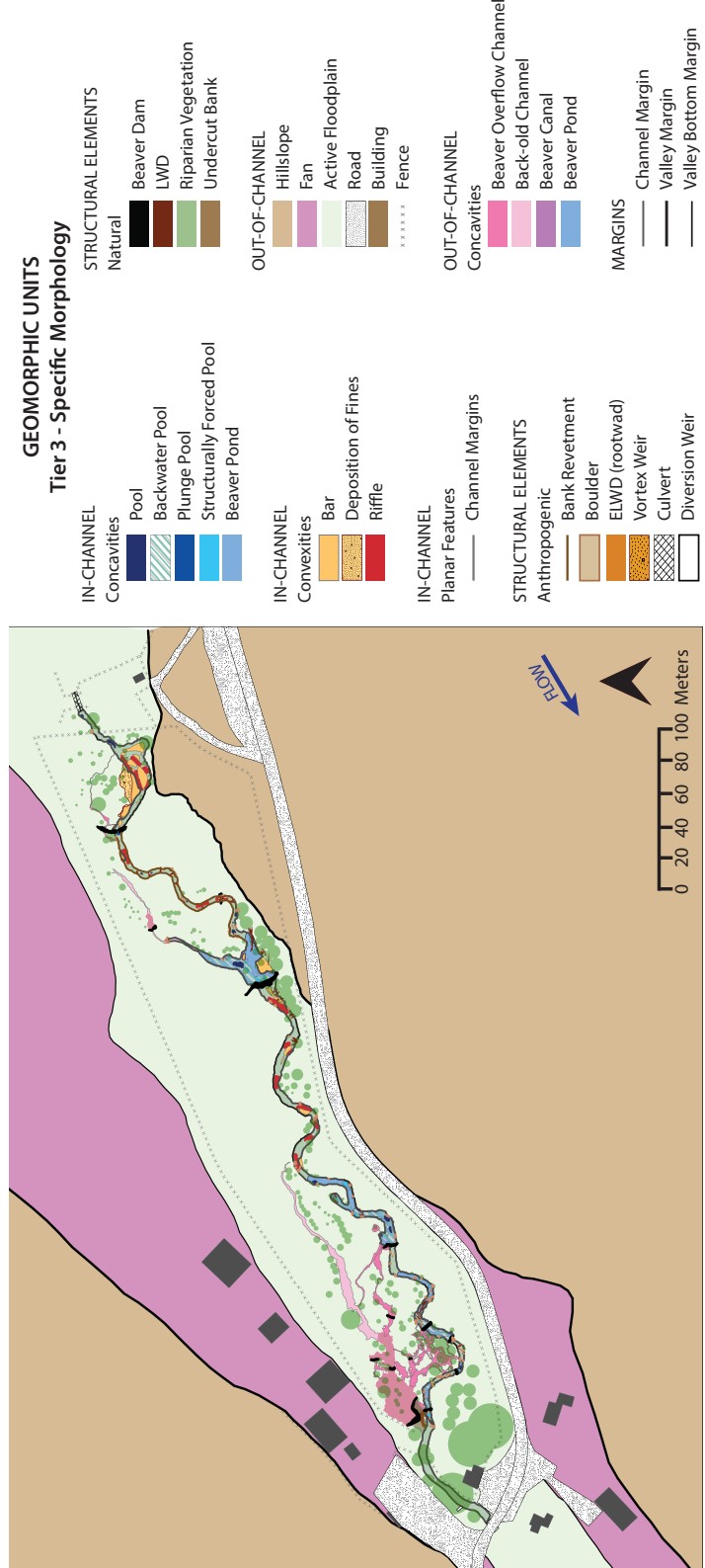

(c) Author(s) 2017. CC BY 4.0 License.



















