# Peer review of "Beaver dam influences on streamflow hydraulic properties and thermal regimes"

_Hydrology and Earth System Sciences, 2017_

## Referee Comment (RC1) · Anonymous Referee #1 · 1 Dec 2017

This manuscript is based on water temperature measurements made at various locations along a stream reach that contains three beaver dam complexes. The temperature measurements cover a three-week period in September, 2013, and are supplemented by modelled depth and velocity distributions generated by application of the Delft3D hydraulic model. The objective is to relate thermal conditions to hydraulic conditions. The analysis consists primarily of a graphical evaluation of the univariate distributions of temperature, velocity and depth, grouped by geomorphic unit class, and time series plots illustrating temperature variability and differences among locations within the reach.

This research is timely and relevant to the readership of HESS given the expansion of beaver populations in many regions and the profound impacts of their activities on

fluvial habitats. However, the data set and the analysis are limited, and the internationally significant contribution of this work is unclear from the current version of the manuscript. Specific comments are provided below. I recommend that the manuscript could be acceptable for publication in HESS following major revisions to address the points raised below.

1. The introduction fails to set up the significance of this work in a convincing manner. For example, after reviewing a number of papers, the authors claim that "[r]egardless of such findings, the connections . . . are not well understood. Many questions remain . . .." Unfortunately, the authors fail to identify any of these questions. In contrast to the authors' statement, I would argue that most if not all of the phenomena noted by the authors at their study site have been documented in previous studies. Furthermore, the physical processes governing thermal variability are well understood in general terms, and most of the observed patterns could be qualitatively deduced from first principles.

Reading between the lines in the introduction, an implicit hypothesis appears to be that the existing literature is not fully relevant to systems with beaver influences. If so, then it would be useful to spell out what is distinctive about beaver-influenced systems. For example, perhaps beaver-affected streams are unique due to the spatial arrangement of the various geomorphic units and the advective connectivity among them.

In any event, I would encourage the authors to identify specific gaps in our current understanding, and then to generate some specific hypotheses or research questions to guide the data analysis, interpretation and presentation (cf McKnight, 2017). Doing so would help to clarify for the reader what this study adds to our understanding, relative to what we have learned from decades of previous stream temperature research.

2. The lack of specificity and clarity noted in the preceding point is exacerbated by issues associated with the use of the literature as supporting information. For example, the authors make a broad statement that "[b]eaver dams and beaver activity can significantly alter . . . various heat exchange mechanisms (e.g.., groundwater exchanges,

Westhoff et al., . . .)." However, none of the cited papers focused on systems influenced by beaver dams, and only Westhoff et al. (2007) specifically quantified the effect of groundwater discharge on stream temperature patterns. Hannah et al. (2004) speculated on the effect of groundwater discharge but made no measurements to quantify it; Keery et al. (2007) described the use of stream and bed temperature time series to compute vertical water exchange between a stream and its bed, but not its influence on stream temperature; and Beschta (1997) did not even mention groundwater.

Another issue related to references is the use of grey-literature and/or perspective articles, such as Sullivan and Adams (1991) and Beschta (1997), and even an AGU abstract (Butler and Hunt, 2013). It would be more effective to draw upon the peer-reviewed and data-based stream temperature literature, and especially papers focused on understanding stream thermal processes and thermal heterogeneity.

3. The analysis is limited to narrative descriptions of univariate distributions of depth, velocity and temperature, grouped by geomorphic units, followed by qualitative interpretations in relation to governing processes. There is no formal statistical analysis and no hypothesis testing (statistical or otherwise), nor is there any formal application of hydraulic or heat-transfer principles to guide the analysis and interpretation. As a consequence, there is no basis for putting the results into a broader context that could be applied to understand or predict thermal heterogeneity for other systems or time periods.

One possible starting point for further analysis is the observation that thermal heterogeneity should depend, in part, on the covariation of depth and velocity among different locations, not just the univariate distributions. It may be useful to examine how the different geomorphic units and their thermal regimes map within a depth-velocity plot, or possibly a space with axes representing non-dimensional numbers relevant to hydraulics and heat transfer (e.g., Reynolds number and possibly the thermal Peclet number).

The authors allude to some heat-budget modelling (line 374ff). I strongly encourage the authors to incorporate that analysis into a revised version of the manuscript.

4. The authors discuss the role of aquatic macrophytes in producing vertical stratification in the backwater areas (line 348ff). Could more information be provided about the macrophytes, such as species, density and height of protrusion above the water surface? Perhaps include a photograph in the supplemental information. Were the macrophytes accounted for in the hydraulic modelling? The authors may find the work of Willis et al. (2017) relevant.

5. The authors refer to local groundwater discharge in the backwater area (line 351). What data were used to determine locations of groundwater discharge? Does groundwater discharge occur at other locations than the backwater area? I suggest the authors add more detail to the site description to address this and the preceding comment.

6. Further to the discussion of stratification in the backwater area, the authors found a time lag of up to 8 hr for the diurnal peak temperature in the bottom layer relative to the surface layer and main channel flow. They also note that Clark et al. found different patterns of time lag. Can the authors suggest a physically based explanation?

7. The authors used a hydraulic model to estimate depth and velocity distributions. They compared simulated depths to measurements and provided values for mean bias error and root-mean-square error. However, no comparisons are reported between modelled and observed velocities. Do the authors have any sense of how large these errors may be, and whether the results are robust to combined errors in depth and velocity?

8. The study focused on a three-week period in September, which was characterized by low-flow conditions. Based on the literature and process considerations, I would expect thermal heterogeneity to increase with decreasing streamflow and with increasing solar radiation. It is unclear whether the observed patterns in this study would

be relevant for other periods, particularly earlier in the summer when both streamflow and solar radiation would be higher. This concern could be addressed to some degree by focusing on the variability of heterogeneity within the study period. (e.g., contrasting day and night, and days with differing insolation).

9. Following on the preceding point, do the authors have meteorological and stream-flow data for the study period? This information would be useful for interpreting the time series plots and the within-period variation of thermal heterogeneity (Figures 5 and 6).

10. The authors relate the observed temperature variability to fish habitat suitability, but only in a passing way (line 359ff). Perhaps this discussion topic could be developed in more detail with reference to the relative roles of temperature and depth/velocity as controls on habitat suitability. For example, environmental flow needs are often deter-mined with respect to depth and velocity, with no reference to thermal regime (Olden and Naiman, 2010). In the current case, do the locations with optimal temperatures coincide with optimal hydraulic conditions for the resident fish species?

11. In Figures 5 and 6, temperature differences are shown between upstream and downstream locations. Examination of the figures suggests that much of the difference is caused by an advective shift in the timing of the diel temperature wave rather than changes in temperature caused by energy exchange (especially Figure 6B). It may be interesting to account for this effect by shifting the time series to account for travel time, and then looking at temperature differences, as was done by Rutherford et al. (2004).

12. In the plots that show upstream-downstream differences as well as temperature time series, it would be useful to adjust the y-axis intervals so that the right-hand axis has a horizontal line corresponding to $\Delta T = 0$ for a visual reference. Indeed, the use of an exaggerated scale for the right-hand y-axis overstates the magnitude of the differ-ences. Consider using the same scale for both temperature series and their difference.

13. The reference list needs editing for completeness and correctness. I have not gone

through the references in detail, but noted the following points:

line 439. "Chapman u. Hall"

line 443. Article title in all caps.

line 445. Issue number missing (each issue is numbered separately).

line 558-559. No journal name or volume number.

References

McKnight, D. M. (2017), Debates - Hypothesis testing in hydrology: A view from the field: The value of hydrologic hypotheses in designing field studies and interpreting the results to advance hydrology, Water Resour. Res., 53, 1779–1783, doi:10.1002/2016WR020050.

Olden, J.D. and R.J. Naiman. (2010). Incorporating thermal regimes into environmental flows assessments: modifying dam operations to restore freshwater ecosystem integrity. Freshwater Biology 55: 86–107.

Rutherford et al. (2004), Effects of patchy shade on stream water temperature: how quickly do small streams heat and cool? Marine and Freshwater Research 55: 737-748.

Willis et al. (2017), Seasonal aquatic macrophytes reduce water temperatures via a riverine canopy in a spring-fed stream. Freshwater Science, DOI: 10.1086/693000.

---

## Referee Comment (RC2) · Anonymous Referee #2 · 13 Dec 2017

This study aims to extend our understanding of the spatial stratification of thermal regimes in streams. It examines the role of geomorphic unit classification and beaver dam complexes on select stream hydraulic properties, which are used to explain observations of high variability in stream thermal regime. To do this, studied was Curtis Creek in northern Utah, which beaver re-colonized in 2009. Concluded in the manuscript was that "geomorphic units within beaver dam complexes exhibit highly unique thermal responses in part due to the variability in flow velocities and depths" (L404). Honestly, this was one of the most clearly written phrases in the manuscript. If true, this manuscript would provide a meaningful contribution to the literature. The problem for me is that I was unable to assess the validity of this conclusion from the material presented. I recommend the authors focus their revisions to the manuscript around this idea, and

describe clear objectives/hypotheses to evaluate/test it. I think much of the density of the manuscript stems from an attempt to capture variations in stream temperature resulting from geomorphic stratification, stream depth and velocity distributions and also beaver dam influences of the geomorphic template – all at varying spatial scales. It caused me to lose focus of the main new insights generated by this study. And, it has led me to recommend rejecting publication of this manuscript in HESS, at least until substantial improvements are made.

Substantial comments: a) The introduction section lacks clear focus and clear questions to drive the research. The authors nicely lay out evidence for spatial stratification in stream temperature regimes. They also argue in the introduction that beaver dams add hydraulic diversity to streams but that their influence on stream temperatures is uncertain as results from previous studies have been contradictory. I was expecting that statement to be followed by some key outstanding questions. Instead, it was simply stated that many questions remain on how to relate different stream temperature responses to varying channel complexity and stream hydraulic properties, especially for streams with beaver living in them. So, I was left confused at the end of the Introduction as to the goal of the study. I ended up thinking that the focus of this manuscript was on characterizing temperature regimes across a range of stream geomorphic units (in a stream that just happens to be beaver habitat), rather than a study of beaver dam influences on stream energetics (as the title suggests). I recommend the authors rethink their focus and write a more compelling introduction that lays out specific gaps in understanding and questions that relate to those gaps. Specially, I think the Introduction needs to be re-worked to more clearly outline how beaver-mediated changes to stream geomorphic structure are unique (or not) from channel complexities found in streams lacking beaver, and why such differences might lead to unexpected impacts on the temperature regime (if that is to be the focus). I do think there is merit in using the geomorphic template to explain thermal heterogeneities in streams, and that such an approach is likely to resolve some of the contradictory findings of the impact of beaver dams that exists in the literature.

b) There is inadequate description of the in-field configuration of the temperature sensors (paragraph starting at L124). Were the loggers placed in some sort of radiation shield to prevent direct solar heating? Were any manual temperature measurements made to ensure solar heating was not occurring? Were the sensors installed at a consistent stream depth? I also think it would be useful to report the expected measurement uncertainties here.

c) How were the simulated depth and velocity distributions (L152) verified and validated? The main conclusions from this manuscript are based on the model producing accurate and credible results.

d) How much difference would an underestimate of stream depth of 0.056 m (from the modeling; L189) likely make to classification of the geomorphic unit type given the small differences in depth thresholds between the classes (classification rules are provided at L165)? I see high potential for mis-classification of geomorphic units. I think disclosure of some of the uncertainties regarding the research design would only serve to strengthen the manuscript. Without such disclosures, it is hard to assess the validity of the conclusions reached.

e) It is useful to present an overall picture of variations in stream temperatures at each geomorphic unit, as was done starting at L258. In addition, I am wondering if there were differences in the diel stream temperature variability among geomorphic units or across the beaver complexes?

f) Section 5.1 – I see this as useful information, but it is not a point worthy of analysis in the overall finding of this research. It is commonly known that small differences in the choice of Manning's n can have a large impact on simulated streamflows. So, I recommend removing this paragraph from the Discussion section, and adding the key components of it into the Methods section.

g) Section 5.2 of the Discussion misses an excellent opportunity, in my view, as it focuses on the least meaningful aspect of the analysis. Instead, what I think is needed

here is a discussion of the how beaver damming impacts the geomorphic classification of the channel, and a linkage of that to stream velocities and depths. Are the changes in channel hydraulics that beaver damming creates well described by geomorphic forms present in channels without beavers? What I am asking is are stream velocity and depth distributions in beaver dammed reaches similar to those in un-dammed reaches, and, how do the differences that occur play out in the thermal regime of the various components of a stream? I think a much more effective Discussion will be guided by a strengthen description of the research goal and objectives in a revised Introduction.

Detailed comments: 1) Plant names should be italicized, for example, willow species should read Salix spp. on L100 with the Salix italicized.

2) Generally, the manuscript needs a thorough grammar edit. Also needed is a consistent style of in-text citation.

3) L217-220: It is cumbersome to read and compare the depth and velocity values for each geomorphic unit as text, given the number of values. I recommend placing these values in a table to facilitate reader understanding.

4) L248: Please provide statistical evidence to support the existence of a warming trend of 1C during the day and a net cooling of 0.5C during the night.

5) L319: Reflects variation of what in the reach – structural unit variations?

6) L340: So, how do DOC concentration and turbidity affect the thermal regime? How important were these factors at Curtis Creek?

---

## Author Comment (AC1) · 24 Jan 2018

Thank you for the Reviewer 1 comments! We responded to individual comments below. The numbering was kept from original RC1.

Reviewer 1 Comments: This manuscript is based on water temperature measurements made at various locations along a stream reach that contains three beaver dam complexes. The temperature measurements cover a three-week period in September, 2013, and are supplemented by modelled depth and velocity distributions generated by application of the Delft3D hydraulic model. The objective is to relate thermal conditions to hydraulic conditions. The analysis consists primarily of a graphical evaluation of the univariate distributions of temperature, velocity and depth, grouped by geomorphic unit

class, and time series plots illustrating temperature variability and differences among locations within the reach. This research is timely and relevant to the readership of HESS given the expansion of beaver populations in many regions and the profound impacts of their activities on fluvial habitats. However, the data set and the analysis are limited, and the internationally significant contribution of this work is unclear from the current version of the manuscript. Specific comments are provided below. I recommend that the manuscript could be acceptable for publication in HESS following major revisions to address the points raised below.

- Thank you for your constructive comments. We believe that the additional statistical analyses described below strengthen the insights gained regarding the relationships between hydraulic properties, geomorphic units, and observed temperature variability (please see #5 below). Further, the extended discussion regarding the implications of this hydraulic and thermal variability on the native fish species provides the links to habitat and the context for these findings (see #13 below).

1. The introduction fails to set up the significance of this work in a convincing manner. For example, after reviewing a number of papers, the authors claim that "[r]egardless of such findings, the connections : : : are not well understood. Many questions remain : : :." Unfortunately, the authors fail to identify any of these questions. In contrast to the authors' statement, I would argue that most if not all of the phenomena noted by the authors at their study site have been documented in previous studies. Furthermore, the physical processes governing thermal variability are well understood in general terms, and most of the observed patterns could be qualitatively deduced from first principles. Reading between the lines in the introduction, an implicit hypothesis appears to be that the existing literature is not fully relevant to systems with beaver influences. If so, then it would be useful to spell out what is distinctive about beaver-influenced systems. For example, perhaps beaver-affected streams are unique due to the spatial arrangement of the various geomorphic units and the advective connectivity among them. In any event, I would encourage the authors to identify specific gaps in our current understanding, and then to generate some specific hypotheses or research questions to guide the data analysis, interpretation and presentation (cf McKnight, 2017). Doing so would help to clarify for the reader what this study adds to our understanding, relative to what we have learned from decades of previous stream temperature research.

- We agree that the introduction did not set up the research questions or literature gaps well and this will be revisited. We will also clarify that we hypothesized that the large temperature variability observed in beaver dam complexes are dictated by the unique distribution of geomorphic units, the corresponding hydraulics (i.e., residence times), and in some locations the hydrology (e.g., local or regional groundwater or hyporheic influences). When you increase the density of specific geomorphic units that are common in beaver impoundments, the residence time distribution changes and the heat transfer or transport roles will shift and alter thermal responses. In the next version of the MS, the introduction will clearly outline our primary objective of determining if the variability in hydraulic characteristics (depth, velocity) associated with the geomorphic units dominant in beaver dam complexes can explain the variability in observed temperature responses. As described below, we will add some multivariate statistical analyses to strengthen the insights gained regarding the relationships between hydraulics, geomorphic classification, and temperature. Further, we will extend the discussion regarding the implications of this hydraulic and thermal variability on native fish species to provide context and a more direct link to expanded habitat.

2. The lack of specificity and clarity noted in the preceding point is exacerbated by issues associated with the use of the literature as supporting information. For example, the authors make a broad statement that "[b]eaver dams and beaver activity can significantly alter : : : various heat exchange mechanisms (e.g.., groundwater exchanges, Westhoff et al., : : :)." However, none of the cited papers focused on systems influenced by beaver dams, and only Westhoff et al. (2007) specifically quantified the effect of groundwater discharge on stream temperature patterns. Hannah et al. (2004) speculated on the effect of groundwater discharge but made no measurements to quantify

it; Keery et al. (2007) described the use of stream and bed temperature time series to compute vertical water exchange between a stream and its bed, but not its influence on stream temperature; and Beschta (1997) did not even mention groundwater.

- It is clear why this sentence created some confusion. The references were meant to be inclusive of the different heat transfer mechanisms identified within the literature that can be influenced by dam construction, but do not discuss these heat fluxes in the context of beaver dams. The "e.g., groundwater exchanges" makes the sentence read as if this is the only mechanism that will change and is misleading. In short, this sentence was meant to be general in nature and not focused on beaver dam impacted streams (in part because there has been limited to no temperature modeling done in such systems). The paragraph will be altered to clarify this point and fit within the more focused introduction.

Another issue related to references is the use of grey-literature and/or perspective articles, such as Sullivan and Adams (1991) and Beschta (1997), and even an AGU abstract (Butler and Hunt, 2013). It would be more effective to draw upon the peer reviewed and data-based stream temperature literature, and especially papers focused on understanding stream thermal processes and thermal heterogeneity.

- Agreed. These will be replaced with peer reviewed references.

3. The analysis is limited to narrative descriptions of univariate distributions of depth, velocity and temperature, grouped by geomorphic units, followed by qualitative interpretations in relation to governing processes. There is no formal statistical analysis and no hypothesis testing (statistical or otherwise), nor is there any formal application of hydraulic or heat-transfer principles to guide the analysis and interpretation. As a consequence, there is no basis for putting the results into a broader context that could be applied to understand or predict thermal heterogeneity for other systems or time periods. One possible starting point for further analysis is the observation that thermal heterogeneity should depend, in part, on the covariation of depth and velocity among
different locations, not just the univariate distributions. It may be useful to examine how the different geomorphic units and their thermal regimes map within a depth-velocity plot, or possibly a space with axes representing non-dimensional numbers relevant to hydraulics and heat transfer (e.g., Reynolds number and possibly the thermal Peclet number).

- We agree a better understanding of the relationship between depth, velocity, geo-morphic classification, and temperature would strengthen the MS. Along these lines, we have done some exploratory analysis with step-wise regression to determine how maximum temperatures and the daily ranges in temperature relate to geomorphic classification, depth, and velocity. In this analysis we found that velocity is not a significant predictor (P>0.1) of stream temperature due to velocities in most of the geomorphic units being similarly low (e.g., pools, margins, backwaters). In contrast, the regression models that include geomorphic unit and depth explain approximately 50% of the variation in maximum stream temperature and approximately 40% of the variation in the daily temperature range. This compares to less than 10% of the variation being explained when only depth and velocity are included for these two models. This suggests that geomorphic classification and geomorphic unit density are important in understanding maximum daily temperatures and daily temperature ranges within beaver dam impacted stream systems. It is clear that the variability in depth and geomorphic units present within beaver dams lead to temperature diversity that supports or fosters growth of aquatic biota.

The authors allude to some heat-budget modelling (line 374ff). I strongly encourage the authors to incorporate that analysis into a revised version of the manuscript.

- While temperature modeling would provide some insight regarding thermal responses, the complexity of the temperatures (as shown within the temperature data) are difficult to capture at the geomorphic scales we are interested in here. The modeling that has been completed for this particular dam accounts for larger scale (but still sub-dam scale) responses and includes many critical heat transfer mechanisms (e.g.,

radiation penetration, bed conduction, etc.) that can be important in these lower velocity variable depth systems. While the Delft modeling captures much of the smaller scale hydraulic complexities (even in 2D simulations), the temperature modeling module within this software is very simplistic and misses many key mechanisms that dictate the thermal variability in such small impoundments. We believe that the introduction of temperature modeling and all the complexities associated with such efforts will detract from the key objectives and findings presented within this paper.

4. The authors discuss the role of aquatic macrophytes in producing vertical stratification in the backwater areas (line 348ff). Could more information be provided about the macrophytes, such as species, density and height of protrusion above the water surface? Perhaps include a photograph in the supplemental information. Were the macrophytes accounted for in the hydraulic modelling? The authors may find the work of Willis et al. (2017) relevant.

- The aquatic vegetation is actually Chara which is an algae species that attach to the sediments via rhizoids and is commonly and incorrectly assumed to be a macrophyte. The average vegetation height was 0.32 m and the average water depth was 0.42 m. This meant that the top layer of water got very warm due to solar radiation and the water within the dense aquatic vegetation stayed cool and created the significant thermal stratification. We will add more specifics regarding the aquatic vegetation within the site description and clarify that this is not a macrophyte. We did not include the vegetation within the hydraulic modeling, however, as discussed within #10 below, this did not significantly impact our predictions due to this being a backwater with very low velocities that supported the growth of this algae species.

5. The authors refer to local groundwater discharge in the backwater area (line 351). What data were used to determine locations of groundwater discharge? Does groundwater discharge occur at other locations than the backwater area? I suggest the authors add more detail to the site description to address this and the preceding comment.

- In Schmadel et al. (2014), groundwater exchanges were established throughout this study reach. Majerova et al. (2015) show the groundwater head gradients near this area to generally be gaining. Further, the northern arm of the beaver dam is connected to "old" channel that only contains groundwater discharge as it is no longer connected to the "new" channel on the surface (MS Figure 1). Both of these studies show that groundwater exchanges can be occurring throughout the reach, but these are variable over time and space. In many places throughout the study reach where velocities are higher, the influences of these exchanges are less apparent due to vertical mixing in the water column. The study site description will be expanded to include a more thorough explanation of groundwater influences in this area.

Majerova, M., Neilson, B. T., Schmadel, N. M., Wheaton, J. M., and Snow, C. J.: Impacts of beaver dams on hydrologic and temperature regimes in a mountain stream, Hydrol. Earth Syst. Sci., 19, 3541-3556, https://doi.org/10.5194/hess-19-3541-2015, 2015.

Schmadel, N. M., Neilson, B. T. and Kasahara, T. (2014), Deducing the spatial variability of exchange within a longitudinal channel water balance. Hydrol. Process., 28: 3088–3103. doi:10.1002/hyp.9854

6. Further to the discussion of stratification in the backwater area, the authors found a time lag of up to 8 hr for the diurnal peak temperature in the bottom layer relative to the surface layer and main channel flow. They also note that Clark et al. found different patterns of time lag. Can the authors suggest a physically based explanation?

- When comparing the temperature of the backwater bottom layer to the main channel flow, the bottom layer temperature is lagged with differences in the temperature range of up to 2 °C. This is mainly due to localized groundwater inputs in the backwater area which are colder, denser, and help keep this area stratified. The presence of aquatic vegetation additionally reduces solar radiation input into the bottom layer and mixing within the water column. The resulting time lag is likely due to these limited advective

and solar radiation influences and leave the bulk of heat transfer to conduction. The time lags in any temperature data within natural systems will be highly dependent on site specific factors like those mentioned above. We will clarify the mechanisms we expect to be creating this lag within the text.

7. The authors used a hydraulic model to estimate depth and velocity distributions. They compared simulated depths to measurements and provided values for mean bias error and root-mean-square error. However, no comparisons are reported between modelled and observed velocities. Do the authors have any sense of how large these errors may be, and whether the results are robust to combined errors in depth and velocity?

- We provided a comparison of the observed and predicted water surface elevations (wsel) from the model calibration in SI Fig. 1. While the focus of the modeling effort was not on model performance per se, we understand the importance of maintaining a good representation of actual conditions. To further test the model abilities, we compared 28 observed and computed velocities within beaver dam complex #1 to show that model predictions are reasonable for this area. We found an RMSE value of 0.065 m/s. The greatest differences between observed and computed values were near the beaver dam structure where flow paths and velocities are mainly influenced by flow through the dam itself and in the inflow area to the pond where a log (not included in the model) is influencing flow velocities locally. As mentioned in the discussion, flows through dams were part of the channel topography, but were maintained via openings in the dam in the model to mimic observed water surface elevations immediately upstream of the structure. This may have led to computational inaccuracies near the dam structures. If these two areas (n = 6) are removed, the RMSE reduces to 0.028 m/s. We will expand the MS to clarify that these comparisons were completed and show that the modeling results were relatively accurate.

8. The study focused on a three-week period in September, which was characterized by low-flow conditions. Based on the literature and process considerations, I would

expect thermal heterogeneity to increase with decreasing streamflow and with increasing solar radiation. It is unclear whether the observed patterns in this study would be relevant for other periods, particularly earlier in the summer when both streamflow and solar radiation would be higher. This concern could be addressed to some degree by focusing on the variability of heterogeneity within the study period. (e.g., contrasting day and night, and days with differing insolation).

- We agree that with a decrease in flow, the thermal heterogeneity among individual geomorphic units within beaver dams would increase. This highlights the importance of these units being present during low summer flows. However, temperature observations (Figure R1) from higher flows (Q = 0.283 cms where the original study was 0.19 cms), still show that thermal variability within each unit (maximum minus minimum temperature for the unit at given time) even though the difference between individual units is not as apparent as the lower flows. Regardless, the trends observed during September low flows were preserved with backwater showing the highest variability and margins and pools following. Even under higher flows, the difference in temperatures above and below the pond (reaching a maximum of 0.5 °C, red line, Figure R1) provides a nice contrast to the thermal variability present within the geomorphic units. In short, the warming effect of the pond and the role of the different geomorphic units are still observable under higher flows. A similar discussion and a plot showing temperature data from this additional time period will be incorporated into the new MS.

9. Following on the preceding point, do the authors have meteorological and streamflow data for the study period? This information would be useful for interpreting the time series plots and the within-period variation of thermal heterogeneity (Figures 5 and 6).

- Thank you for the comment. We will add following figure (Figure R2) showing air temperature, solar radiation, and stream flow to the Supplemental Information.

10. The authors relate the observed temperature variability to fish habitat suitability, but

only in a passing way (line 359ff). Perhaps this discussion topic could be developed in more detail with reference to the relative roles of temperature and depth/velocity as controls on habitat suitability. For example, environmental flow needs are often determined with respect to depth and velocity, with no reference to thermal regime (Olden and Naiman, 2010). In the current case, do the locations with optimal temperatures coincide with optimal hydraulic conditions for the resident fish species?

- We agree with the reviewers that assessing the fish habitat suitability in relationship to temperature would benefit this manuscript. Many papers that explore the effects of beaver ponds on stream temperature and fish have focused on their positive or detrimental effects on maximum stream temperature (Collen and Gibson 2001). Our findings within this paper indicate that the presence of beaver ponds increases the density of a subset of geomorphic units which then increases the thermal, depth, and velocity variability. In our revised manuscript we will seek to better portray how the observed 15oC daily temperature variability (or range in observed temperature differences) within a beaver dam occurs across the geomorphic units assessed. In making this argument, we will demonstrate that this increased variability provides optimal thermal habitat for cutthroat trout (the native fish species of interest) at all times even if temperatures in some areas are sub-optimal (Figure R3). Furthermore, we will expand this discussion to argue that although temperature in some of the habitat units may not have been thermally suitable during the time this beaver dam complex was instrumented, the units showing warm day temperatures during summer/early fall may be critical to providing additional time for growth during the spring and fall (Olden and Naiman 2010). We will make a similar argument when it comes to areas with lower flow velocities. These beaver ponds provide ideal conditions during winter as the low flow velocity minimizes metabolic cost. In the spring when cutthroat spawn, they can find refuge in the low velocities areas, but have nearby access to flow velocities that are suitable for building redds and laying eggs. We will additionally discuss how these temperature distributions differ as you remove the geomorphic unit density created within the beaver dam complex.

11. In Figures 5 and 6, temperature differences are shown between upstream and downstream locations. Examination of the figures suggests that much of the difference is caused by an advective shift in the timing of the diel temperature wave rather than changes in temperature caused by energy exchange (especially Figure 6B). It may be interesting to account for this effect by shifting the time series to account for travel time, and then looking at temperature differences, as was done by Rutherford et al. (2004).

- We are assuming that the reviewer was referring to Figure 5B. This comment presents an interesting idea, however, the travel time is only 43 minutes based on the mean velocity of 0.25 m/s. As shown in the plots below (Figure R4), the lag in temperatures upstream and downstream is due in part to advective influences (or primarily advective influences during some days). When applying the shift (see bottom subplot), the rising limb of the temperatures are aligned, but for many days, a small lag remains in the falling limb. As you mention, this suggests that the difference in temperatures are due to the actual heat exchanges that occur over the study reach. This can also be seen due in the slightly higher max temperatures and higher average temperatures at the downstream end. With this shift applied, the rate of warming is the same at both locations and they reach the maximum temperature at approximately the same time. However, the downstream temperature sensors is 0.38 °C warmer on average with the largest consistent differences occurring during the peak and times of cooling. Based on some calculations from the modeling results in Stout et al. (2017), we found that the total water volume/surface area in the reach feeding the upstream sensor was half that of the beaver dam impacted reach shown here that influences the downstream sensor. This may suggest that solar radiation (that can penetrate the water column) will warm different depth waters at a similar rate, however, the surface heat fluxes responsible for cooling (sensible, radiative, and latent heat fluxes) will cool deeper portions of the system more slowly.

Stout T.L., Majerova M., Neilson, B.T.: Impacts of beaver dams on channel hydraulics and substrate characteristics in a mountain stream, Ecohydrology, 2017, 10:e1767,

doi: 10.1002/eco.1767,

12. In the plots that show upstream-downstream differences as well as temperature time series, it would be useful to adjust the y-axis intervals so that the right-hand axis has a horizontal line corresponding to _T = 0 for a visual reference. Indeed, the use of an exaggerated scale for the right-hand y-axis overstates the magnitude of the differences. Consider using the same scale for both temperature series and their difference.

- Thank you, the figure was adjusted as suggested.

13. The reference list needs editing for completeness and correctness. I have not gone through the references in detail, but noted the following points: line 439. "Chapman u. Hall", line 443. Article title in all caps, line 445. Issue number missing (each issue is numbered separately), line 558-559. No journal name or volume number.

- Thank you for catching this. We will ensure that the bibliography that is produced by the software is complete.

————————- Figure captions for Figures 1R-4R attached separately:

Figure R1 Similar to Figure 6 in the MS, the temperature difference (maxima minus minima) for individual geomorphic units within beaver dam complex #1 demonstrates the thermal variability created by the beaver dam. The seven-day period shown captures higher flows of 0.29 m2/s. Lines represent temperature variation within pools (solid light blue), backwater (dashed dark blue), and marginal areas (dotted yellow). The dashed red line illustrates the influence of the beaver pond by showing differences between temperature below and above the pond with positive values demonstrating downstream warming effect.

Figure R2 Air temperature, solar radiation, and discharge for study period.

Figure R3 Distribution of temperatures recorded in the warmest of each of the four habitat units on the warmest day during the study period. The added solid red line represents the suitability of stream temperature for cutthroat trout growth where the

[Figure]

apex is the most suitable.

Figure R4 Figure 6b from the original MS (top) and the data from Figure 6b with the upstream data shifted 40 minutes to account for travel time (bottom).
* * *
[Figure]

[Figure]

**Fig. 1.**

[Figure]

Fig. 2.

Figure R3

[Figure]

**Fig. 3.**

[Figure]

**Fig. 4.**

---

## Author Comment (AC2) · 24 Jan 2018

Thank you for the Reviewer 2 comments! We responded to individual comments below. The numbering of the comments follows the original RC2.

Reviewer 2 Comments:

This study aims to extend our understanding of the spatial stratification of thermal regimes in streams. It examines the role of geomorphic unit classification and beaver dam complexes on select stream hydraulic properties, which are used to explain observations of high variability in stream thermal regime. To do this, studied was Curtis Creek in northern Utah, which beaver re-colonized in 2009. Concluded in the manuscript was that "geomorphic units within beaver dam complexes exhibit highly unique thermal re-

sponses in part due to the variability in flow velocities and depths" (L404). Honestly, this was one of the most clearly written phrases in the manuscript. If true, this manuscript would provide a meaningful contribution to the literature. The problem for me is that I was unable to assess the validity of this conclusion from the material presented. I recommend the authors focus their revisions to the manuscript around this idea, and describe clear objectives/hypotheses to evaluate/test it. I think much of the density of the manuscript stems from an attempt to capture variations in stream temperature resulting from geomorphic stratification, stream depth and velocity distributions and also beaver dam influences of the geomorphic template – all at varying spatial scales. It caused me to lose focus of the main new insights generated by this study. And, it has led me to recommend rejecting publication of this manuscript in HESS, at least until substantial improvements are made.

- Thank you for the constructive comment. Your summary of the intended objectives within this paper are correct. Our primary objective is to determine if the variability in hydraulic characteristics (depth, velocity) associated with the dominant geomorphic units in beaver dam complexes can explain the variability in observed temperature responses. The original introduction did not clearly outline the knowledge gap or state these objectives and needs to be clarified. Beyond the original more qualitative measures provided in the manuscript, as discussed below, we will add some multivariate statistical analyses to strengthen the insights gained regarding the relationships between hydraulic, geomorphic classification, and observed temperature responses. Further, we will extend the discussion regarding the implications of this hydraulic and thermal variability on native fish species to provide context and a more direct link to expanded habitat.

Substantial comments: a) The introduction section lacks clear focus and clear questions to drive the research. The authors nicely lay out evidence for spatial stratification in stream temperature regimes. They also argue in the introduction that beaver dams add hydraulic diversity to streams but that their influence on stream temperatures is

uncertain as results from previous studies have been contradictory. I was expecting that statement to be followed by some key outstanding questions. Instead, it was simply stated that many questions remain on how to relate different stream temperature responses to varying channel complexity and stream hydraulic properties, especially for streams with beaver living in them. So, I was left confused at the end of the Introduction as to the goal of the study. I ended up thinking that the focus of this manuscript was on characterizing temperature regimes across a range of stream geomorphic units (in a stream that just happens to be beaver habitat), rather than a study of beaver dam influences on stream energetics (as the title suggests). I recommend the authors rethink their focus and write a more compelling introduction that lays out specific gaps in understanding and questions that relate to those gaps. Specially, I think the Introduction needs to be re-worked to more clearly outline how beaver-mediated changes to stream geomorphic structure are unique (or not) from channel complexities found in streams lacking beaver, and why such differences might lead to unexpected impacts on the temperature regime (if that is to be the focus). I do think there is merit in using the geomorphic template to explain thermal heterogeneities in streams, and that such an approach is likely to resolve some of the contradictory findings of the impact of beaver dams that exists in the literature.

- Thank you for this constructive comment. As stated above, we agree that the introduction needs to be revisited to clearly articulate our primary objectives. We can also see that a new title could be developed that highlights the key contributions. Something along the lines of "Linking beaver dam geomorphic complexity and hydraulic characteristics to thermal variability." We also agree that this approach will help explain some of the contradictions present within the literature.

b) There is inadequate description of the in-field configuration of the temperature sensors (paragraph starting at L124). Were the loggers placed in some sort of radiation shield to prevent direct solar heating? Were any manual temperature measurements made to ensure solar heating was not occurring? Were the sensors installed at a

consistent stream depth? I also think it would be useful to report the expected measurement uncertainties here.

- We will provide more detail regarding the temperature sensor deployment and sensor accuracy/resolution in this section of the MS. In short, all sensors were covered in aluminum foil to avoid radiation influences. All sensors were placed in the center of the water column, with the exception of the vertical arrays where they were spaced evenly throughout the water column. All sensors were placed in temperature baths regularly to ensure that the sensors were reporting values within the expected error of +/-0.2 C.

c) How were the simulated depth and velocity distributions (L152) verified and validated? The main conclusions from this manuscript are based on the model producing accurate and credible results.

- We provided a comparison of the observed and predicted water surface elevations (wsel) from the model calibration in SI Fig. 1. While the focus of the modeling effort was not on model performance per se, we understand the importance of maintaining a good representation of actual conditions. To further test the model abilities, we compared 28 observed and computed velocities within beaver dam complex #1 to show that model predictions are reasonable for this area. We found an RMSE value of 0.065 m/s. The greatest differences between observed and computed values were near the beaver dam structure where flow paths and velocities are mainly influenced by flow through the dam itself and in the inflow area to the dam where a log (not included in the model) is influencing flow velocities. As mentioned in the discussion, flow through dams were part of the channel topography, but were maintained via openings in the dam in the modeling to mimic observed water surface elevations immediately upstream of the structure. This may have led to computational inaccuracies near the dam structures. If these two areas (n = 6) are removed, the RMSE reduces to 0.028 m/s. We will expand the MS to clarify that these comparisons were completed and show that the modeling results were relatively accurate.

d) How much difference would an underestimate of stream depth of 0.056 m (from the modeling; L189) likely make to classification of the geomorphic unit type given the small differences in depth thresholds between the classes (classification rules are provided at L165)? I see high potential for mis-classification of geomorphic units. I think disclosure of some of the uncertainties regarding the research design would only serve to strengthen the manuscript. Without such disclosures, it is hard to assess the validity of the conclusions reached.

- We agree that the error in depth predictions could be acknowledged. While depth thresholds for units have small differences, velocity was used at the same time. If we vary thresholds by 10% and look at predicted average depth for each unit, we found that pool classification was most affected. If the lower threshold is varied from 0.45m and 0.55 m (from 0.5 m), the average predicted depth varies between 0.62 and 0.69 m (original prediction for pools was 0.66 m). In terms of pool area, when computation cell coverage is used, for example increasing the threshold to 0.55 m would mean decrease in pool area about 20 %, which translates to change of overall pool representation for the entire reach from 13 % to 10 %. Riffles and margins exhibit no significant change when their upper thresholds are moved by 10%, and the average depth holds at 0.13 m and 0.06 m, respectively. The backwater unit is also not affected by model under- or over-predictions as it covers the entire depth range of 0.03 m to the maximum depth. Even though pool boundaries would slightly change with threshold change, this would not affect the results regarding the geomorphic units and temperature as the temperature sensors were mainly placed in the middle of the geomorphic units.

e) It is useful to present an overall picture of variations in stream temperatures at each geomorphic unit, as was done starting at L258. In addition, I am wondering if there were differences in the diel stream temperature variability among geomorphic units or across the beaver complexes?

- This comment is a bit unclear. Based on our understanding, we have included these data and figures in the current manuscript. For example, Figure 6 in the MS shows

the range of temperatures experienced within each geomorphic unit (and therefore between geomorphic units) as well as the differences in temperature across the entire pond (shown in red dashed line). Figure 5 in the MS shows the actual temperatures and temperature differences across the beaver dam complex and over the entire study reach.

f) Section 5.1 – I see this as useful information, but it is not a point worthy of analysis in the overall finding of this research. It is commonly known that small differences in the choice of Manning's n can have a large impact on simulated streamflows. So, I recommend removing this paragraph from the Discussion section, and adding the key components of it into the Methods section.

- We agree. It is commonly known that small differences in the choice of Manning's n can have a large impact on simulated streamflows. We will remove this paragraph and add the key points to the methods.

g) Section 5.2 of the Discussion misses an excellent opportunity, in my view, as it focuses on the least meaningful aspect of the analysis. Instead, what I think is needed here is a discussion of the how beaver damming impacts the geomorphic classification of the channel, and a linkage of that to stream velocities and depths. Are the changes in channel hydraulics that beaver damming creates well described by geomorphic forms present in channels without beavers? What I am asking is are stream velocity and depth distributions in beaver dammed reaches similar to those in un-dammed reaches, and, how do the differences that occur play out in the thermal regime of the various components of a stream? I think a much more effective Discussion will be guided by a strengthen description of the research goal and objectives in a revised Introduction.

- Thank you for very helpful and constructive comment. We will expand the discussion to clarify the role of beaver dams on shifting the geomorphic variability based on our field/model based mapping efforts. We will look at the total area of each geomorphic unit in the portion of the study reach where beaver dams are present and compare

it sections without beaver dams. Further, we will clarify how the findings of Stout et al. (2017) tie into this paper by discussing how the velocity and depth distributions differ between beaver impacted and non-impacted reaches as well as pre and post pre-beaver colonization. While Stout et al. differs in that a 1D hydraulic model was applied and the geomorphic unit level information is not captured, the reach depth and velocity distributions offer meaningful information on how dam complexes alter stream channel hydraulics and therefore, influence thermal responses.

Detailed comments: 1. Plant names should be italicized, for example, willow species should read Salix spp. on L100 with the Salix italicized.

- We will change this in the new version of the MS.

2. Generally, the manuscript needs a thorough grammar edit. Also needed is a consistent style of in-text citation.

- As we incorporate these edits, we will ensure consistency in citations and in writing.

3. L217-220: It is cumbersome to read and compare the depth and velocity values for each geomorphic unit as text, given the number of values. I recommend placing these values in a table to facilitate reader understanding.

- The attached table will be added to the manuscript (Table 1R).

4. L248: Please provide statistical evidence to support the existence of a warming trend of 1C during the day and a net cooling of 0.5C during the night.

- The statistics will be included in the manuscript and are provided in the attached table (Table 2R).

5. L319: Reflects variation of what in the reach – structural unit variations?

- The sentence will be changed to:

"The range of reach scale temperatures reflect temperature variability within the reach

(Fig. 5A) and highlights the warming effects of a series of beaver complexes on longitudinal stream temperature patterns (Fig. 5B)."

6. L340: So, how do DOC concentration and turbidity affect the thermal regime? How important were these factors at Curtis Creek?

- Turbidity and DOC concentrations are extraordinarily low in Curtis Creek. Therefore, these are not factors that need to be considered here. This has been clarified by changing the sentence to:

"While both depth and velocities within pools are key to quantifying thermal stratification in Curtis Creek, other factors such as dissolved organic carbon and turbidity must also be considered in some systems. . ."
* * *
*Table 1R  Model calculated depths and velocities for individual geomorphic units.*

| | Depth (m) | | | Velocity (m s$^{-1}$) | | |
|---|---|---|---|---|---|---|
| | Average | Min | Max | Average | Min | Max |
| Pools | 0.660 | 0.500 | 1.080 | 0.110 | 0.001 | 0.730 |
| Backwaters | 0.380 | 0.030 | 1.080 | 0.030 | 0.000 | 0.100 |
| Margins | 0.060 | 0.030 | 0.100 | 0.030 | 0.000 | 0.100 |
| Riffles | 0.130 | 0.030 | 0.400 | 0.640 | 0.002 | 1.830 |

**Fig. 1.**

*Table 1R Statistics for downstream net warming during the day and net cooling during the night over the study reach (Fig. 5B) and beaver dam complex #1 (Fig. 5D).*

|  | Study reach | | Beaver Dam Complex #1 | |
| --- | --- | --- | --- | --- |
|  | Warming (°C) | Cooling (°C) | Warming (°C) | Cooling (°C) |
| Min | 0.024 | -0.530 | 0.024 | -0.655 |
| Max | 1.275 | -0.024 | 1.432 | -0.024 |
| Mean | 0.458 | -0.147 | 0.227 | -0.099 |
| Standard Deviation | 0.256 | 0.109 | 0.292 | 0.132 |

**Fig. 2.**